# The impact of tinnitus on adult cochlear implant recipients: A mixed-method approach

Kelly K. S. Assouly[1,2,3]*, Maryam Shabbir[4], Bas van Dijk[3], Derek J. Hoare[4,5], Michael A. Akeroyd[4,5], Robert J. Stokroos[1,2], Inge Stegeman[1,2], Adriana L. Smit[1,2]

1 Department of Otorhinolaryngology and Head & Neck Surgery, University Medical Center Utrecht, Utrecht, the Netherlands, 2 UMC Utrecht Brain Center, University Medical Center Utrecht, Utrecht University, Utrecht, the Netherlands, 3 Cochlear Technology Centre, Mechelen, Belgium, 4 Hearing Sciences, Mental Health and Clinical Neurosciences, School of Medicine, University of Nottingham, Nottingham, United Kingdom, 5 NIHR Nottingham Biomedical Research Centre, Nottingham, United Kingdom

* k.k.s.assouly@umcutrecht.nl

## Abstract

### Background

Tinnitus is a common problem in patients with a cochlear implant (CI). Between 4% and 25% of CI recipients experience a moderate to severe tinnitus handicap. However, apart from handicap scores, little is known about the real-life impact tinnitus has on those with CIs. We aimed to explore the impact of tinnitus on adult CI recipients, situations impacting tinnitus, tinnitus-related difficulties and their management strategies, using an exploratory sequential mixed-method approach.

### Methods

A 2-week web-based forum was conducted using Cochlear Ltd.'s online platform, Cochlear Conversation. A thematic analysis was conducted on the data from the forum discussion to develop key themes and sub-themes. To quantify themes and sub-themes identified, a survey was developed in English with face validity using cognitive interviews, then translated into French, German and Dutch and disseminated on the Cochlear Conversation platform, in six countries (Australia, France, Germany, New Zealand, the Netherlands and United Kingdom). Participants were adult CI recipients experiencing tinnitus who received a Cochlear Ltd. CI after 18 years of age.

### Results

Four key themes were identified using thematic analysis of the discussion forum: tinnitus experience, situations impacting tinnitus, difficulties associated with tinnitus and tinnitus management. Among the 414 participants of the survey, tinnitus burden on average was a moderate problem without their sound processor and not a problem with the sound processor on. Fatigue, stress, concentration, group conversation and hearing difficulties were the most frequently reported difficulties and was reported to intensify when not wearing the sound processor. For most CI recipients, tinnitus seemed to increase when performing a hearing test, during a CI programming session, or when tired, stressed, or sick. To manage

**Data Availability Statement:** All relevant data are within the paper and its Supporting Information files.

**Funding:** KKSA and MS received funding from the European Union's Horizon 2020 research and innovation program under the Marie Sklodowska-Curie grant (grant number 764604). DH and MAA are funded through the National Institute for Health and Care Research (NIHR) Biomedical Research Centre programme. The funders had no role in study design, data collection and analysis, decision to publish, or preparation of the manuscript.

**Competing interests:** KKSA and MS received funding from the European Union's Horizon 2020 research and innovation program under the Marie Sklodowska-Curie grant (agreement number 764604). KSSA and BvD are employed at Cochlear Technology Centre, Mechelen, Belgium. The content of the study belongs to the authors alone and do not reflect Cochlear Technology Centre policy. No further conflict of interest is reported by the authors. This does not alter our adherence to PLOS ONE policies on sharing data and materials.

their tinnitus, participants reported turning on their sound processor and avoiding noisy environments.

## Conclusion

The qualitative analysis showed that tinnitus can affect everyday life of CI recipients in various ways and highlighted the heterogeneity in their tinnitus experiences. The survey findings extended this to show that tinnitus impact, related difficulties, and management strategies often depend on sound processor use. This exploratory sequential mixed-method study provided a better understanding of the potential benefits of sound processor use, and thus of intracochlear electrical stimulation, on the impact of tinnitus.

## Introduction

Tinnitus is the perception of sound in the ears or in the head without an external stimulus, often described as ringing or buzzing in the ears [1]. It can vary in sound qualities and location and can also impact people differently. Some experience tinnitus as not bothersome at all, while others experience it as bothersome and debilitating, resulting in a substantial reduction in quality of life [2]. The prevalence of chronic tinnitus ranges between 10 and 15% in the general adult population and is higher among hearing impaired patients [3–5], with up to 80% of tinnitus prevalence in cochlear implant candidates [6].

For those severely affected by hearing loss, a cochlear implant (CI) could be considered, to restore speech perception function. The CI primarily aims to restore hearing by providing electrical stimulation to the auditory nerve. While the primary purpose of cochlear implantation is to restore hearing, systematic reviews showed that tinnitus reduction can be a secondary beneficial effect [7–9]. However, the effect of implantation seems to vary widely between patients, ranging from total tinnitus suppression to tinnitus induction in up to 9% of CI users [6,10]. Recent studies compared tinnitus presence pre- and post-implantation in patients receiving a cochlear implant for severe to profound hearing and showed that tinnitus prevalence decreased post-implantation [10–13]. In these studies, between 34% and 53% of cochlear implant (CI) users still experienced tinnitus after implantation [10–13] and only between 4% and 25% experienced it as a moderate to severe handicap [14–16].

There is a gap in the literature regarding the impact of tinnitus on CI users' everyday life. In the general population experiencing tinnitus, it can result in substantial reductions in quality of life and associated emotional and functional difficulties [2]. In CI users, tinnitus could be influenced by the intracochlear electrical stimulation provided by the sound processor, which has shown potential to reduce tinnitus [17]. As shown in studies assessing the effect of CI on tinnitus, the presence and impact of tinnitus seems to differ significantly when wearing the processor or not wearing the processor in CI users experiencing tinnitus [9,18,19]. Several tinnitus questionnaires have been developed and validated to assess different aspects of tinnitus impact in the general population. However, these questionnaires do not necessarily address the complexity of tinnitus for CI users.

To better understand how tinnitus impacts CI users in their everyday life, we adopted an exploratory sequential mixed-method approach [20]. Using a qualitative phenomenological approach, we aimed to explore the impact of tinnitus on CI users, the difficulties associated with tinnitus, the situations impacting tinnitus and the tinnitus management strategies. We then developed a survey to quantify how tinnitus impacts CI users, what difficulties are

associated with tinnitus, what situations impact tinnitus, and how they manage it. To investigate the influence of the sound processor on tinnitus, a secondary aim was to assess the presence of tinnitus with regards to the sound processor status in this cohort of cochlear implant users.

## Materials and methods

### Study design

This observational study is based on a mixed-method approach consisting of two parts: (1) an exploratory sequential design involving collecting qualitative exploratory data using a phenomenological approach, and (2) using the findings to develop a survey for CI users to quantitatively measure the impact of tinnitus in CI users.

For the purpose of this study, a web-based approach was used to collect qualitatively rich data for a large and diverse pool of CI users [21]. Cochlear Conversation is a web-based platform designed by Cochlear Ltd., offering several discussion forums and surveys on topics related to their CI or bone conduction devices. Members of the platform are Cochlear™ Nucleus®, BAHA® and OSIA® system users who have agreed to the terms and conditions of Cochlear Conversation.

Although a theoretical framework was not explicitly defined a priori, the authors considered the ESIT Framework of variables defining and characterizing tinnitus sub-phenotypes particularly relevant to the current work as it describes the high dimensionality of tinnitus heterogeneity [22]. We followed the Mixed Methods Article Reporting Standards (MMARS) for reporting this study [23].

### Ethical considerations

The Central Committee on Research Involving Human Subjects in the Netherlands (CCMO) confirmed that the Medical Research Involving Human Subjects Act did not apply to the study. Therefore, an ethical waiver was obtained. This study was performed according to the declaration of Helsinki. The participants provided informed consent to participate in the forum discussion and the survey and to use their data after anonymization, in compliance with the General Data Protection Regulation (GDPR).

### Part 1: Forum study

**Forum design.**   The forum study relied on prospectively gathered data from the Cochlear Conversation web-based platform. The discussion forum was launched on the Cochlear Conversation platform in four European countries: France, Germany, United Kingdom, and the Netherlands. Registered users of the Cochlear Nucleus implant were invited by email to participate in the discussion forum. The invitation and reminder emails contained a link to the discussion forum website. Participation was on a voluntary basis and all posts were submitted anonymously.

The forum discussion was a moderator-led online forum discussion where a moderator encouraged the discussion and participants discussed specific topics through posting a series of messages and commented on each other's post [24]. Every 2 days, the forum opened the discussion, per country, on pre-defined topics: impact of tinnitus on everyday life ('How much are you affected by your tinnitus?',' In what situations does tinnitus most impact your everyday life?'), impact of CI on tinnitus ('How did receiving your cochlear implant affect your tinnitus experience?') and management strategies used to manage tinnitus ('What do you do to manage your tinnitus?') (Fig 1). Each forum per country was moderated by an independent moderator

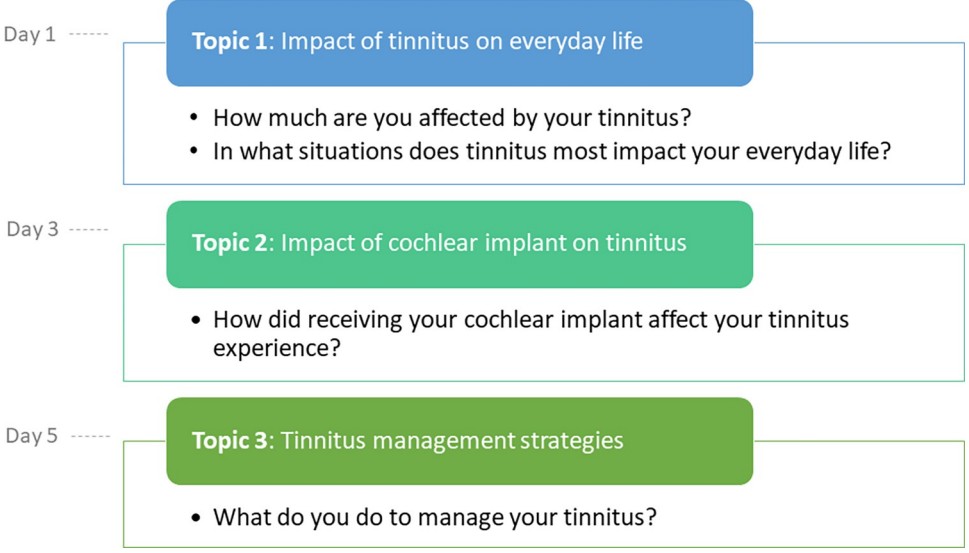

**Fig 1. Design of the web-based discussion forum.** The forum opened the discussion on a new topic every two days. Each topic was open for discussion until the forum closed, six days after it opened.

using a flexible moderation style. All moderators were trained by the researcher KKSA about the phenomenological approach and research objectives. Depending on the content of their post, the moderator encouraged participants to elaborate their responses by referring to a pre-defined script, containing a series of questions. Pseudonyms rather than names were used to distinguish individuals, preserving their anonymity.

**Forum participants.** *Recruitment.* Participants were adult CI recipients who received a Cochlear Nucleus implant (Cochlear Ltd., Macquarie University, NSW, Australia) and were registered on the Cochlear Conversation platform. They were invited via email to voluntarily participate in the discussion forum, with the single criterion that they had tinnitus. The level of tinnitus burden patients experience was not used as an inclusion criterion. Before receiving access to the discussion forum, participants provided digital informed consent, agreeing that their data would be anonymized and used for research purposes.

*Sample size and participant characteristics.* Of the 222 participants who consented to participate in this forum study, 136 submitted one or more posts on the discussion forum (55 from France, 33 from Germany, 14 from the Netherlands and 34 from the United Kingdom). The remaining 86 participants joined the forum but did not post in the discussion. To preserve anonymity, we did not collect further demographic information on the participants.

**Qualitative analysis.** A qualitative analysis was performed based on the responses from the forum. Responses were clustered into categories using the inductive thematic analysis method described by Braun and Clarke [25]. Authors KKSA and MS familiarized themselves with the messages on the forum by reading and re-reading the data. Codes were then individually created by highlighting manually and making notes on key findings for each of the discussion topic questions. Codes were then refined, by a process of merging and adding to the initial codes. Finally, the codes were discussed between KKSA and MS and any discrepancies were resolved to improve the descriptions of the codes. This was followed by grouping codes with related topic together into categories, developing sub-themes, and finally, into themes using mind maps to illustrate the relationship between the different themes. The themes were discussed and a final version of the coding manual was agreed upon by the two coders to reflect the tinnitus experiences of the CI users from the data set.

## Part 2: Survey study

**Survey development.** A survey in English was developed to quantify the themes: tinnitus impact in adult CI recipients, situations impacting tinnitus, tinnitus-related difficulties, and management strategies. The first step of the survey development was the creation of the list of sub-themes emerging from the qualitative analysis of the forum discussion. The number of questions in the survey was determined based on the number of sub-themes derived from the descriptive quantitative analysis. To build the survey content, validated tinnitus questionnaires (Tinnitus Functional Index [26], Tinnitus Handicap Questionnaire [27], Self-Efficacy for Tinnitus Management Questionnaire [28], Sound Sensitive Tinnitus Index [29], Tinnitus Cognitions Questionnaire [30], Tinnitus Reaction Questionnaire [31], Tinnitus Handicap Inventory [32], Tinnitus Questionnaire [33], Tinnitus Primary Function Questionnaire [34], Tinnitus Magnitude Index [35], International Tinnitus Inventory [36], Tinnitus Coping Style Questionnaire [37], Fear of Tinnitus Questionnaire [38], Chronic Tinnitus Acceptance Questionnaire [39], Subjective Tinnitus Severity Scale [40], Tinnitus Acceptance Questionnaire [41], Tinnitus Fear Avoidance Scale [42]) were reviewed to extract available questions which were judged to be related to the identified sub-themes. After reviewing the lists of available questions by sub-themes, two authors (KKSA and MS) chose which question to use for each sub-theme based on their own judgement. Where no question was available to assess a sub-theme, a question was developed.

KKSA and MS refined the survey to harmonize the structure and order of the questions and response options. KKSA and MS also added additional response options to existing questions: "other, please give details", "don't know" and "none" in case none of the existing response options were judged to be relevant for the participants. The final version of the survey was presented to the rest of the research team and the content and structure was discussed and further refined based on their input. Any changes required were made after a consensus with all authors. This was done iteratively until a final version was approved.

**Survey validity.** As the survey contained novel questions and response options, face validity needed to be confirmed. This was done using cognitive interviews. A cognitive interview is a one-on-one interview, exploring how respondents process information as they complete a questionnaire, detecting problems that respondents may have in understanding survey instructions and in providing responses [43]. For this, CI users registered as volunteers of Cochlear Benelux were invited by email to participate in the cognitive interviews. Participation was on a voluntary basis and the only condition to participate was to experience tinnitus. Before the interview, a list of probing questions was prepared by KKSA. The interview probe is available in the publicly available dataset. During the interview, participants were invited to verbalize their mental process involved in providing responses to each question. The methodology used for the cognitive interviews followed the guidelines by ISPOR [44].

KKSA conducted two cognitive interviews with each of two volunteer CI users experiencing tinnitus. The interviewer, KKSA, had extensive experience of conducting interviews as part of her doctoral studies and had no prior relationship to the interviewees. She carried out the interviews and took notes during and after the interviews. The interviews were not recorded. No personal information was collected during the interviews. The interviews were conducted online with participant verbal consent and lasted between 30 to 60 minutes. The survey was modified based on the first cognitive interviews with each participant after which KKSA conducted a second round with the same participants to validate the changes made and agree on the new version of the survey. The validation of the new version and the absence of further comments from the two participants did not require another round. After validation, the survey was translated from English to French, German and Dutch by external translators, verified

for cultural appropriateness by native speakers [45] and disseminated on the Cochlear Conversation platform.

**Survey dissemination.** Patients willing to participate were asked to complete the 15-minute survey containing questions about their current tinnitus experience, situations impacting tinnitus, difficulties associated with tinnitus, impact of CI on their tinnitus burden and their management strategies. Patient characteristics gender, age range, laterality of implantation, bimodal hearing, and type of hearing loss were collected. No personal identifiable information was collected from the survey. The survey was open from 8 November 2021 to 5 December 2021 and implemented on the Cochlear Conversation platform in the official language of the country in Australia, France, Germany, New Zealand, the Netherlands, and United Kingdom.

**Survey participants.** *Recruitment.* Participants were adult CI recipients who received a Cochlear™ Nucleus® implant (Cochlear Ltd., Macquarie University, NSW, Australia) and were registered on the Cochlear Conversation platform. They were invited via email to voluntarily participate in the online survey, with the criterion that they currently experience tinnitus. Participants who had already taken part in the discussion forum could also take part in the survey. Therefore, the survey population might, to some extent, overlap with the one of the discussion forum. Before participating, they digitally signed an informed consent agreeing that their data will be anonymized and used for research purposes. A reminder email was also sent 1 week after the first invitation to try maximizing participation.

*Sample size and participant characteristics.* Four-hundred and fourteen participants completed the survey across six countries, Australia: $n = 104$, France: $n = 65$, Germany: $n = 167$, New Zealand: $n = 16$, the Netherlands: $n = 29$, United Kingdom: $n = 33$. About eighty percent ($n = 329$) of the participants were aged between 65 and 84 years of age, and 54.8% ($n = 227$) of the participants were female (Table 1). Participants had different hearing profiles, with 87.9% reporting bilateral hearing loss (*hearing loss in both ears equally*: 43.0% ($n = 178$); *hearing loss in both ears, but in one more than in other*: 44.9% ($n = 186$)) and 12.1% ($n = 50$) reporting

**Table 1. Patient demographics.**

| Demographics | N (%) |
|---|---|
| Age | |
| *19–24* | 4 (1.0) |
| *25–49* | 62 (15.0) |
| *50–64* | 146 (35.3) |
| *65–84* | 183 (44.2) |
| *85+* | 5 (1.2) |
| *Missing* | 14 (3.4) |
| Gender | |
| *Male* | 187 (45.2) |
| *Female* | 227 (54.8) |
| Hearing loss type | |
| *Hearing loss in one ear, good hearing in the other ear* | 50 (12.1) |
| *Hearing loss in both ears equally* | 178 (43.0) |
| *Hearing loss in both ears, but in one more than in other* | 186 (44.9) |
| Device use | |
| *Unilateral CI stimulation* | 111 (26.8) |
| *Bilateral CI stimulation* | 121 (29.2) |
| *Bimodal stimulation* | 182 (44.0) |
| Hours wearing the SP per day | |
| *Median (IQR)* | 15 (13–16) |
| *Range* | 0–24 |

IQR: Interquartile; N: Number of CI recipients; SP: Sound processor.

unilateral deafness (Table 1). Forty-four percent (*n* = 182) were bimodal users, using a CI on one side and a hearing aid on the other side, 26.8% (*n* = 111) were unilateral CI recipients and 29.2% (*n* = 121) were bilaterally implanted (Table 1). They reported wearing their sound processor on average 15.0 hours per day (IQR: 13–16, Table 1).

## Data analysis

Descriptive statistics were performed on the dataset. Proportion were used for categorical or binary variables and median and interquartile (IQR) were used for continuous variables. The main study parameters were the survey outcomes on the following items: tinnitus presence, tinnitus impact, tinnitus-related difficulties, situations impacting tinnitus and management strategies use.

Tinnitus presence was measured by a multiple-choice question (Q1, S1 Table). The effect of CI on tinnitus was measured by two multiple choice questions (Q6-7, S1 Table). Tinnitus awareness and annoyance were rated on a numerical scale ranging between 0 "none of the time" to 100 "all of the time" (Q9-10, S1 Table).

Tinnitus impact was measured by three multiple choice questions assessing different conditions: in general, when wearing the sound processor, and when not wearing the sound processor. Each question was rated using five impact levels ranging between "not a problem" to "a very big problem" (Q11-13, S1 Table).

Difficulties were assessed by 13 questions using visual analogue scales (Q14-26, S1 Table). Participants were asked to focus on the difficulties caused by tinnitus, independent of difficulties caused by hearing loss. Each difficulty was rated in two conditions: when wearing the sound processor and when not wearing the sound processor. The rating scale ranged from 0 for "never" to 10 for "always".

Situations impacting tinnitus were assessed by three multiple choice questions using visual analogue scales (Q27-29, S1 Table). The first question (Q27, S1 Table) assessed 10 situations rated on a numerical scale between 0 "increase tinnitus" and 10 "reduce tinnitus", with 5 corresponding to "no change". The ratings were then grouped into three categories: *increase tinnitus* for all the ratings between 0 and 4, *decrease tinnitus* for all the ratings between 6 and 10 and *no change* for all the ratings of 5. Other situations were depicted by recipients using open field texts (Q28-29, S1 Table) and were then grouped into similar themes.

Management strategies use and effect were measured by four multiple choice questions (Q30-33, S1 Table). Participants were asked how easy it is to manage tinnitus in general on a numerical scale between 0 "very easy" and 10 "impossible" (Q33, S1 Table). Responses provided in the open field text were grouped into similar themes.

The secondary outcome measure was the presence of tinnitus depending on the use of the sound processor, which relied on a multiple-choice question (Q1, S1 Table). Fisher tests were performed to assess the difference in the occurrence of difficulties between the two conditions: when wearing the sound processor and when not wearing it. All analyses were performed in R Studio 1.3.1073 (®R Studio). A *p*-value lower than 0.05 indicated a statistically significant result.

## Results

### Part 1: Forum study

**Themes derived from forum data.** Four key themes were identified from the thematic analysis of the forum: (1) *tinnitus experience*, (2) *situations impacting tinnitus*, (3) *difficulties associated with tinnitus* and (4) *tinnitus management strategies*. The sub-themes and codes

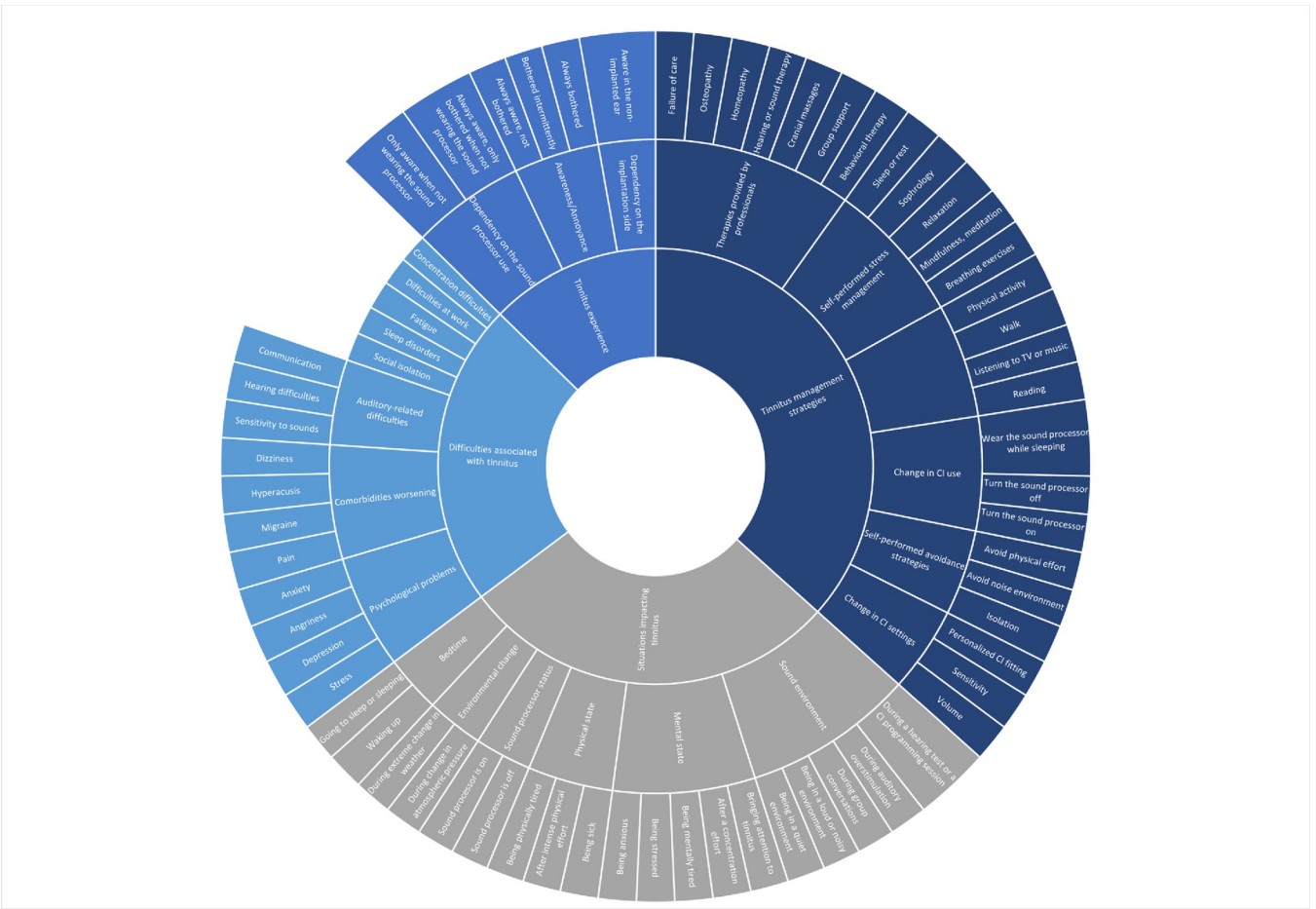

**Fig 2. Themes, sub-themes, and codes from the thematic analysis.**

emerging from the discussion thread under each main theme are presented in Fig 2 and summarized in S2 Table.

*Theme 1*: *Tinnitus experience.* Different degrees of tinnitus awareness and annoyance were reported by the participants (Fig 2). Some participants described the tinnitus as "*a friend*" or "*a music in the head*" which they are aware of without any associated burden. Whereas others characterized their tinnitus as "*uncomfortable*", "*unbearable*" or "*a problem*". Participants reported that tinnitus awareness and annoyance can depend on the sound processor use (*always aware*, *only bothered when not wearing the sound processor* and *only aware when not wearing the sound processor*, Fig 2): "*No tinnitus when I have my implant, they are present just after I have unhinged when I go to bed*". According to the participants, tinnitus awareness can also depend on which side was implanted with the CI. Some participants reported tinnitus only on the non-implanted side (*aware in the non-implanted ear*, Fig 2): "*Since implantation on the left ear, almost more tinnitus on the right*".

*Theme 2*: *Situations impacting tinnitus.* Overall, six sub-themes related to the key theme *situations impacting tinnitus* emerged from the thematic analysis: *bedtime*, *environmental change*, *mental state*, *physical state*, *sound environment* and *sound processor status* (Fig 2). At *bedtime*, CI users felt an increase in tinnitus (*when going to sleep*, *when sleeping* or *when waking up*, Fig 2): "*I feel a lot of tinnitus especially before going to bed and getting up in the morning*". Participants mentioned that changes in their environment affected their tinnitus (*extreme change in*

*weather* or *change in atmospheric pressure*, Fig 2): "*Stress, noise and, suddenly, extreme changes in weather make tinnitus worse*".

Mental states relating to some form of emotional distress such as *being anxious*, *stressed*, *mentally tired* or *after a concentration effort* were answered to have an impact on their tinnitus: "*tinnitus occurs when fatigue occurs*". For some participants, focusing on their tinnitus experience made their tinnitus worse (*when bringing attention to tinnitus*, Fig 2): "*The more focused you are on your tinnitus the louder the sound gets*". Physical states related to *intense physical effort*, *being physically tired* or *sick* were also listed as situations having an impact on their tinnitus: "*When I jog, I remove my processors and the intense effort created tinnitus*".

Participants said that tinnitus impact can vary depending on the sound environment. Depending on the individual, tinnitus impact was reported to be worse in presence of sounds (*in loud or noisy environment*, *during group conversations* or during *auditory overstimulation*, Fig 2): "*tinnitus has developed mainly in a noisy atmosphere*" or in absence of sounds (*in quiet environment*, Fig 2): "*In situations of silence, my tinnitus reappears despite the activated processors*". CI users experiencing tinnitus also mentioned *hearing test* or *CI programming session* as situations that could affect tinnitus impact: "*I have to say that it is enormously strong every time I do hearing tests, specifically the one with the beep. Afterwards my head is buzzing, and I can almost only hear noises.*".

CI users reported tinnitus being more present or bothersome when they were not wearing their sound processor (*when the sound processor is off*, Fig 2) and therefore noticed a change in tinnitus impact when they were wearing their sound processor (*when the sound processor is on*, Fig 2): "*Decrease sharply when my implant is activated, always present when I remove my external processor*".

*Theme 3*: *Difficulties associated with tinnitus*. Participants noted that tinnitus can, to various degrees, cause difficulties in different aspects of their daily life (Fig 2). In addition to hearing loss, *auditory-related problems* appeared to be associated with tinnitus (*hearing difficulties*, *communication difficulties*, *sensitivity to sounds*, Fig 2): "tinnitus bothers me to hear and listen to ambient sounds". Other comorbidities such as *dizziness*, *hyperacusis*, *migraine* and *pain* were described as created or exacerbated by tinnitus: "*I had to take off the CI as soon as I noticed that the tinnitus was getting stronger. If I didn't do that, dizziness and pain followed, and hearing was especially painful in the high notes.*".

Participants described *concentration difficulties* resulting in *fatigue*: "*This comes with a lot of fatigue in my daily life.*" as well as *difficulties at work*: "*In online meetings the concentration is lost if tinnitus dominates too much, they happen to me to refuse encounters that would put me in difficulty because too embarrassed*". Participants also mentioned *fatigue* and *sleep disorders* related to their tinnitus: "*At night it just wakes me up and makes sleep difficult again.*" Psychological problems such as *anxiety*, *anger*, *depression*, and *stress* developed or were aggravated by tinnitus: "*I am continually anxious and stressed by this tinnitus.*". Participants who reported situations that could make their tinnitus worse described *social isolation* due to avoiding such situations: "*Sometimes I isolate myself so that I can be operational again.*".

*Theme 4*: *Tinnitus management strategies*. Participants discussed their ways of managing tinnitus, which included strategies varying between self-administrated practices, such as *stress management* or *distraction activities*, to *therapies provided by professionals* (Fig 2).

As CI users, participants reported *turning on* or *off their sound processor* and *changing their CI settings* (*changing volume*, *changing sensitivity*, *having personalized CI fitting*, Fig 2) to manage their tinnitus depending on the situation: "*I mostly unplug my implant and prosthesis when I'm overworked by tinnitus*", "*I had to take my remote control and lower the sensitivity or volume to support*". Discussion on tinnitus management revealed that different strategies were used depending on the time of the day, during the day or at night, and whether they were wearing

their sound processor: "*During the day I can largely ignore the tinnitus—it's always there in the background but I am now used to the continual sound, although I would prefer not to have it. Initially at night the tinnitus prevented me from sleeping but I began streaming sounds through my processor using the GN Resound App.*".

Recipients were able to ignore their tinnitus, although aware of it under certain conditions: "*Mostly I am able to ignore it.*". Most participants said that CI provided sufficient improvement to their tinnitus and don't need further strategies: "*Since having the implant I have simply found that wearing it helps dull the tinnitus and make it more bearable.*". Some participants explained that they tried to avoid situations or environments where tinnitus can get worse (*isolation, avoid noisy environment, avoid physical effort*, Fig 2): "*I try to be careful to avoid all that is acoustic disturbances that can amplify tinnitus in the left ear*". Other participants mentioned managing their tinnitus by performing activities to distract themselves or manage their stress: "*You have to be able to live with them, to forget them. Daily physical activity (to aerate the brain, to cause physical fatigue that helps you fall asleep). Have a playful intellectual activity that occupies the brain.*".

Some participants reported attending *therapies provided by professionals*, whereas others said they never tried tinnitus therapies: "*I'm afraid I haven't managed to find any other remedies for my tinnitus. I have never been offered any tinnitus therapy*". Some participants reported following *behavioral therapy* or *group support* to learn to deal with their tinnitus: "*I made an initial appointment with a behavior therapist in my area. [. . .] During the therapy I found out together with the therapist what stresses me and how I can deal with it better. [. . .] Over a longer period of time, I have learned to recognize stressful situations early on and to deal with them better. [. . .] In the event of setbacks, the therapist supported me wonderfully and helped me to keep going.*". Participants mentioned the effect of *cranial massages* in reducing tinnitus-related distress "*I had a session of cranial massages which was very relaxing and learnt to where to run my fingers to replicate a near massage. It did relax as I mentioned and sleep was easier, although the Tinnitus remained.*". *Hearing or sound therapy* were reported by participants: "*Initially at night the tinnitus prevented me from sleeping but I began streaming sounds through my processor using the GN Resound App*". Alternatives therapies such as *homeopathy* or *osteopathy* were also mentioned by participants: "*it's osteopathy that's helping me most right now by discouraging my necks and jaws, which are very tense.*". *Failure of care* was also mentioned by participants in the forum: "*I've tried all the usual recommendations such as listening to music & other recorded sounds, meditation, mindfulness, exercise. None really help.*"

## Part 2: Survey study

**Tinnitus characteristics.** Table 2 summarizes tinnitus characteristics of the study participants. The presence of tinnitus did not depend on the sound processor use for 68.4% (*n* = 283) of the participants. Among those reporting tinnitus depending on the sound processor use, 29.7% (*n* = 123) had tinnitus only when they were not wearing their sound processor and 1.9% (*n* = 8) had it only when wearing the sound processor. Tinnitus was described as constant by 54.8% (*n* = 227) of participants and intermittent by the other 45.2% (*n* = 187). Tinnitus was unilateral for 24.7% (*n* = 102) of participants (*right ear*: 10.2% (*n* = 42); *left ear*: 14.0% (*n* = 58); *sometimes on the left, sometimes on the right*: 0.5% (*n* = 2)), bilateral for 45.2% (*n* = 166) of participants (*both ears but worse in the right*: 17.4% (*n* = 72); *both ears but worse in the left*: 22.7% (*n* = 94); *both ears equally*: 5.1% (*n* = 21)) and inside the head for 28.0% (*n* = 116) of them (*inside the head*: 25.8% (*n* = 107); *both ears and inside the head*: 1.2% (*n* = 5); *somewhere specific in the head*: 1.0% (*n* = 4)). Participants were aware of their tinnitus 30% (IQR: 10–70) of their time awake and were annoyed on average 20% (IQR: 10–50) of their time awake.

**Table 2. Tinnitus characteristics.**

| Tinnitus characteristics | N (%) |
|---|---|
| Tinnitus presence | |
| *It does not depend on my SP use* | 283 (68.4) |
| *Only when I am wearing my SP* | 8 (1.9) |
| *Only when I am not wearing my SP* | 123 (29.7) |
| Tinnitus type | |
| *Constant*: I can always or usually hear it | 227 (54.8) |
| *Intermittent*: "comes and goes" | 187 (45.2) |
| Tinnitus side | |
| *Right ear* | 42 (10.2) |
| *Left ear* | 58 (14.0) |
| *Both ears but worse in the right* | 72 (17.4) |
| *Both ears but worse in the left* | 94 (22.7) |
| *Inside the head* | 107 (25.8) |
| *Somewhere else* | 33 (8.0) |
| *Both ears equally* * | 21 (5.1) |
| *Both ears and inside the head* * | 5 (1.2) |
| *Sometimes on the left, sometimes on the right* * | 2 (0.5) |
| *Surrounding me* * | 1 (0.2) |
| *Somewhere specific in the head* * | 4 (1.0) |
| *Don't know* | 8 (1.9) |
| Tinnitus awareness | |
| *Median (IQR)* | 30 (10–70) |
| *Range* | 0–100 |
| Tinnitus annoyance | |
| *Median (IQR)* | 20 (10–50) |
| *Range* | 0–100 |
| Tinnitus pre-implantation | |
| *Yes* | 341 (82.4) |
| *No* | 58 (14.0) |
| *Don't know* | 15 (3.6) |
| Tinnitus changes post-implantation | |
| *Yes, my tinnitus got better while wearing my SP* | 154 (37.2) |
| *Yes, my tinnitus got better while not wearing my SP* | 6 (1.5) |
| *Yes, my tinnitus got worse while wearing my SP* | 20 (4.8) |
| *Yes, my tinnitus got worse while not wearing my SP* | 36 (8.7) |
| *Yes, I got tinnitus after my implantation* | 43 (10.4) |
| *Yes, I no longer have tinnitus* | 10 (2.4) |
| *Yes, other change* | 51 (12.3) |
| *Tinnitus sounds changed* * | 11 (2.7) |
| *Tinnitus got better* * | 10 (2.4) |
| *Tinnitus got worse* * | 10 (2.4) |
| *Tinnitus got worse after implantation and then got better* * | 4 (1.0) |
| *Better in one side and worse in the other side* * | 3 (0.7) |
| *Tinnitus side changed* * | 3 (0.7) |
| *Sometimes better, sometimes worse* * | 3 (0.7) |
| *Other* * | 7 (1.7) |
| *No* | 122 (29.5) |
| *Don't know* | 29 (7.0) |

IQR: Interquartile; N: Number of CI recipients; SP: Sound processor. Options marked with an asterisk (*) are extracted from an open field question and grouped into similar themes.

Fourteen percent ($n = 58$) of recipients reported not experiencing tinnitus prior to implantation, 82.4% ($n = 341$) reported having tinnitus pre-implantation and 3.6% ($n = 15$) did not know. No change in tinnitus was noticed between pre and post implantation for 29.5% ($n = 122$) of the recipients and 7.0% ($n = 29$) did not recall change. The other 63.5% ($n = 263$) of recipients reported tinnitus changes post-implantation. Positive changes described by

recipients could depend on the sound processor use (*tinnitus got better while wearing the sound processor*: 37.2% (*n* = 154); *tinnitus got better while not wearing the sound processor*: 1.5% (*n* = 6)) or not (*no longer have tinnitus*: 2.4% (*n* = 10); *tinnitus got better*: 2.4% (*n* = 10)). Recipients also accounted for negative changes post-implantation (*tinnitus got worse while wearing the sound processor*: 4.8% (*n* = 20); *tinnitus got worse while not wearing the sound processor*: 8.7% (*n* = 87); *got tinnitus after implantation*: 10.4% (*n* = 43); *tinnitus got worse*: 2.4% (*n* = 10)). Changes in tinnitus sounds (2.7%, *n* = 11), changes in tinnitus side (0.7%, *n* = 3) or fluctuating changes (*tinnitus got worse after implantation and then got better*: 1.0% (*n* = 4); *better in one side and worse in the other side*: 0.7% (*n* = 3); *sometimes better, sometimes worse*: 0.7% (*n* = 3)) were also described by participants.

## Tinnitus impact

Outcomes of pre-implantation tinnitus impact and post-implantation tinnitus impact evaluated when wearing the sound processor, when not wearing the sound processor, and in general are shown in Table 3.

Most CI recipients (63.9%, *n* = 218) described their pre-implantation tinnitus as a moderate or a big problem (*moderate problem*: 39.6% (*n* = 135); *a big problem*: 24.3% (*n* = 83)) and 15.8% (*n* = 54) described their tinnitus as a very big problem. In general, post-implantation, a small problem (25.4%, *n* = 105) and a moderate problem (36.5%, *n* = 151) were reported, with 10.1% (*n* = 42) reporting their tinnitus as being not a problem and 7.5% (*n* = 31) reporting it as a very big problem.

When wearing the sound processor, most recipients rated their tinnitus as not a problem (31.9%, *n* = 132) or a small problem (30.9%, *n* = 128), and 10.2% rated it as a big to very big problem (*big problem*: 8.0% (*n* = 33); *very big problem*: 2.2%, *n* = 9). When not wearing the sound processor, most recipients reported having a moderate problem (35.0%, *n* = 145) or a big problem (23.4%, *n* = 97), and a minority qualified their tinnitus as not a problem (6.8%, *n* = 28). There was a statistically significant difference in tinnitus impact between wearing the sound processor and not wearing it (Chi square test, $X^2$ = 202.75, *p* <0.01).

**Situations impacting tinnitus.** Table 4 summarizes the occurrence and rating of situations identified as impacting tinnitus. The most frequently scored situations impacting tinnitus negatively were *when stressed* (90.3% (*n* = 374); described in open field text: 12.3% (*n* = 51)), *when tired* (90.8% (*n* = 376); described in open field text: 2.9% (*n* = 12)), *when sick* (92.5% (*n* = 383)) and *during a hearing test or CI programming session* (90.8% (*n* = 376)). Moreover, the situations *when waking up*, *when sick*, and *during a hearing test or CI programming session* had the lowest ratings, with a median of 0.0 (IQR: 0.0–3.0)). With the exception of *when anxious*, which was categorized as not impacting tinnitus by 35.3% (*n* = 146) of participants, all

**Table 3. Tinnitus impact.**

| Tinnitus impact N (%) | Pre-implantation (*n* = 341) | Post-implantation *General* (*n* = 414) | Post-implantation *With SP* (*n* = 414) | Post-implantation *Without SP* (*n* = 414) |
|---|---|---|---|---|
| *Not a problem* | 10 (2.9) | 42 (10.1) | 132 (31.9) | 28 (6.8) |
| *A small problem* | 59 (17.3) | 105 (25.4) | 128 (30.9) | 95 (23.0) |
| *A moderate problem* | 135 (39.6) | 151 (36.5) | 112 (27.0) | 145 (35.0) |
| *A big problem* | 83 (24.3) | 85 (20.5) | 33 (8.0) | 97 (23.4) |
| *A very big problem* | 54 (15.8) | 31 (7.5) | 9 (2.2) | 49 (11.8) |

N: Number of CI recipients; SP: Sound processor.

**Table 4. Situations impacting tinnitus.**

| Situations affecting tinnitus | N (%) | Median (IQR) |
|---|---|---|
| When you wake up | | 0.0 (0.0–3.0) |
| *Increase tinnitus* | 350 (84.5) | 0.0 (0.0–1.0) |
| *Decrease tinnitus* | 64 (15.6) | 9.0 (8.0–10.0) |
| *No change* | 0 (0.0) | |
| When you are tired | | 1.0 (0.0–3.0) |
| *Increase tinnitus* | 376 (90.8) | 1.0 (0.0–3.0) |
| *Decrease tinnitus* | 38 (9.2) | 8.0 (7.0–9.0) |
| *No change* | 0 (0.0) | |
| When you are going to sleep | | 1.0 (0.0–3.0) |
| *Increase tinnitus* | 361 (87.2) | 0.0 (0.0–2.0) |
| *Decrease tinnitus* | 53 (12.8) | 8.0 (7.0–9.0) |
| *No change* | 0 (0.0) | |
| When you are in a quiet environment | | 1.0 (0.0–3.0) |
| *Increase tinnitus* | 363 (87.7) | 1.0 (0.0–2.5) |
| *Decrease tinnitus* | 51 (12.3) | 8.0 (7.5–10.0) |
| *No change* | 0 (0.0) | |
| When you are in a loud environment | | 1.0 (0.0–4.0) |
| *Increase tinnitus* | 317 (76.6) | 0.0 (0.0–2.0) |
| *Decrease tinnitus* | 97 (23.4) | 8.0 (7.0–9.0) |
| *No change* | 0 (0.0) | |
| When you are performing a hearing test or CI programming session | | 0.0 (0.0–3.0) |
| *Increase tinnitus* | 376 (90.8) | 0.0 (0.0–2.0) |
| *Decrease tinnitus* | 38 (9.2) | 8.0 (7.0–9.0) |
| *No change* | 0 (0.0) | |
| After physical effort | | 1.0 (0.0–3.0) |
| *Increase tinnitus* | 366 (88.4) | 0.0 (0.0–2.0) |
| *Decrease tinnitus* | 48 (11.6) | 8.0 (7.0–9.0) |
| *No change* | 0 (0.0) | |
| When you are sick | | 0.0 (0.0–3.0) |
| *Increase tinnitus* | 383 (92.5) | 0.0 (0.0–2.0) |
| *Decrease tinnitus* | 31 (7.5) | 8.0 (7.0–9.5) |
| *No change* | 0 (0.0) | |
| When you are stressed | | 1.0 (0.0–3.0) |
| *Increase tinnitus* | 374 (90.3) | 0.0 (0.0–2.0) |
| *Decrease tinnitus* | 40 (9.7) | 7.0 (6.0–9.0) |
| *No change* | 0 (0.0) | |
| When you are anxious | | 4.0 (1.0–5.0) |
| *Increase tinnitus* | 230 (55.5) | 2.0 (0.0–3.0) |
| *Decrease tinnitus* | 38 (9.2) | 8.0 (7.0–9.8) |
| *No change* | 146 (35.3) | |
| Other situations where tinnitus gets better | 148 (35.7) | |
| *When wearing the SP (and hearing aid)* * | 37 (8.9) | |
| *When listening music or other auditory input* * | 24 (5.8) | |
| *When being distracted* * | 21 (5.1) | |
| *When being relaxed, not stressed* * | 19 (4.6) | |
| *When being in a quiet environment* * | 9 (2.2) | |
| *When being at rest* * | 8 (1.9) | |
| *After physical exercises* * | 7 (1.7) | |
| *Other situations* * | 19 (4.6) | |

(*Continued*)

**Table 4.** (Continued)

| Situations affecting tinnitus | N (%) | Median (IQR) |
|---|---|---|
| Other situations where tinnitus gets worse | 182 (44.0) | |
| *When being stressed* * | 51 (12.3) | |
| *When being in loud/noisy/crowed environment* * | 36 (8.7) | |
| *When not wearing the SP* * | 22 (5.3) | |
| *When being in a quiet environment* * | 15 (3.6) | |
| *When being tired* * | 12 (2.9) | |
| *When concentrated or doing a mental/listening effort* * | 11 (2.7) | |
| *When being concerned* * | 10 (2.4) | |
| *When drinking alcohol or coffee* * | 7 (1.7) | |
| *After physical exercises* * | 6 (1.4) | |
| *When falling asleep* * | 6 (1.4) | |
| *Other situations* * | 46 (11.1) | |

IQR: Interquartile; CI: Cochlear implant; N: Number of CI recipients; SP: Sound processor.

Each situation was rated according to the scale: 0 Increases tinnitus– 1–2–3–4–5 No change– 6–7–8–9–10 Reduces tinnitus.

The *Increase tinnitus* group consists of participants rated the situation between 0 and 4.

The *Decrease tinnitus* group consists of participants rated the situation between 6 and 10.

The *No change* group corresponds to participants rated the situation equal to 5.

Options marked with an asterisk (*) are extracted from an open field question and grouped into similar themes.

situations were ranked as negatively impacting tinnitus by at least 84.5% participants. For less than a quarter of participants, the most common situations impacting tinnitus positively were *when being in a loud environment* (23.4% ($n = 97$)), *when being in a quiet environment* (12.3% ($n = 51$); described in open field text: 2.2% ($n = 9$)) and *when waking up* (15.6% ($n = 64$)).

**Tinnitus-related difficulties.** Table 5 summarizes the ratings of participants on the occurrence of 12 tinnitus-related difficulties. Fatigue, group conversation and hearing difficulties were the most frequently reported difficulties when wearing the sound processor, with a median score of 2 out of 10 (*fatigue*: 2.0 (IQR: 0.0–4.0); *group conversation*: 2.0 (IQR: 0.0–5.0); *hearing difficulties*: 2.0 (IQR: 0.0–5.0)). Without sound processor, group conversation and hearing difficulties were the most frequently reported, with a median score of 4, followed by difficulties in listening to radio or TV, concentration difficulties and stress, with a median score of 3. All tinnitus-related difficulties were significantly more present when not wearing the sound processor (Fisher's exact test, $p < 0.001$). Some tinnitus-related difficulties, such as sleep disorders, depressive feeling, anxiety, anger, and difficulties at work, were on average never present while wearing the sound processor (*sleep disorders*: 0.0 (IQR: 0.0–3.0), depressive feeling: 0.0 (IQR: 0.0–2.0), anxiety: 0.0 (IQR: 0.0–2.0), anger: 0.0 (IQR: 0.0–2.8), difficulties at work: 0.0 (IQR: 0.0–3.0)) but appeared when not wearing it (*sleep disorders*: 2.0 (IQR: 0.0–6.0), depressive feeling:: 1.0 (IQR: 0.0–4.0), anxiety: 1.0 (IQR: 0.0–4.0), anger: 1.0 (IQR: 0.0–4.0), difficulties at work: 1.0 (IQR: 0.0–5.0)). For fatigue, the lowest difference in occurrence between the two conditions was shown, with and without sound processor.

Eighty out of 414 participants (19.3%) mentioned other health problems caused or aggravated by tinnitus (Q26, S1 Table). These comorbidities were extracted from an open field question and grouped into similar themes (S3 Table). Balance disorders ($n = 12$, 2.9%), depression ($n = 10$, 2.4%), migraine ($n = 11$, 2.7%), hypertension ($n = 6$, 1.4%) and neck pain ($n = 6$, 1.4%) were the most mentioned comorbidities. Although already rated in the previous questions, anxiety ($n = 6$, 1.4%), concentration difficulties ($n = 3$, 0.7%), hearing difficulties ($n = 2$, 0.5%), fatigue ($n = 10$, 2.4%), sleep disorders ($n = 8$, 1.9%) and stress ($n = 11$, 2.7) were also mentioned.

**Table 5. Tinnitus-related difficulties with and without sound processor (SP) on.**

| Difficulties | Median (IQR) | p-value |
|---|---|---|
| Sleep disorders | | **<0.001** |
| *With SP* | 0.0 (0.0–3.0) | |
| *Without SP* | 2.0 (0.0–6.0) | |
| Fatigue | | **<0.001** |
| *With SP* | 2.0 (0.0–4.0) | |
| *Without SP* | 2.0 (0.0–7.0) | |
| Stress | | **<0.001** |
| *With SP* | 1.0 (0.0–4.8) | |
| *Without SP* | 3.0 (1.0–7.0) | |
| Depressive feeling | | **<0.001** |
| *With SP* | 0.0 (0.0–2.0) | |
| *Without SP* | 1.0 (0.0–4.0) | |
| Anxiety | | **<0.001** |
| *With SP* | 0.0 (0.0–2.0) | |
| *Without SP* | 1.0 (0.0–4.0) | |
| Anger | | **<0.001** |
| *With SP* | 0.0 (0.0–2.8) | |
| *Without SP* | 1.0 (0.0–4.0) | |
| Concentration | | **<0.001** |
| *With SP* | 1.0 (0.0–4.0) | |
| *Without SP* | 3.0 (1.0–7.0) | |
| Work | | **<0.001** |
| *With SP* | 0.0 (0.0–3.0) | |
| *Without SP* | 1.0 (0.0–5.0) | |
| Hearing | | **<0.001** |
| *With SP* | 2.0 (0.0–5.0) | |
| *Without SP* | 4.0 (0.0–9.0) | |
| Listening radio/TV | | **<0.001** |
| *With SP* | 1.0 (0.0–5.0) | |
| *Without SP* | 3.0 (0.0–8.0) | |
| Group conversation | | **<0.001** |
| *With SP* | 2.0 (0.0–5.0) | |
| *Without SP* | 4.0 (0.0–10.00) | |
| Social life | | **<0.001** |
| *With SP* | 1.0 (0.0–4.0) | |
| *Without SP* | 2.0 (0.0–8.0) | |

IQR: Interquartile; N: Number of CI recipients; SP: Sound processor; TV: Television.

Each difficulty was rated according to the scale: 0 Never– 1–2–3–4–5–6–7–8–9–10 Always.

P-values are from Fisher's exact tests between the two conditions: With SP and without SP. Bold indicates statistically significant p<0.05.

**Tinnitus management strategies.** Tinnitus management strategies and techniques adopted by CI recipients are shown in Table 6. They described tinnitus being easier to manage when wearing the sound processor compared to when not wearing it (*with sound processor*: 2.0 (IQR: 0.0–4.0); *without sound processor*: 5.0 (IQR: 2.0–8.0)). Turning on their sound processor was the most successful and used tinnitus strategy during the day, as rated by 32.1% (*n* = 133). Additionally, 8.5% (n = 35) of recipients changed their CI settings. On the other hand, only 8.7% (*n* = 36) turned off their sound processor as a management strategy. To manage their tinnitus during the day, 26.8% (*n* = 111) avoided noisy situations and 20.4% (*n* = 84) avoided silent situations. Activities such physical exercises (25.3%, *n* = 105) and relaxing activities (24.2%, *n* = 101) were rated as improving tinnitus. Finally, 18.9% (*n* = 78) did not use any specific management strategy.

**Table 6. Tinnitus management strategies and techniques.**

| Tinnitus management | N (%) | Median (IQR) |
|---|---|---|
| **Tinnitus management level** | | |
| With SP | 414 (100.0) | 2.0 (0.0–4.0) |
| Without SP | 414 (100.0) | 5.0 (2.0–8.0) |
| **Day management strategies/techniques** | | |
| Avoid noisy situations | 111 (26.8) | 7.0 (5.0–9.0) |
| Avoid silent situations | 84 (20.4) | 7.0 (6.0–9.0) |
| Physical exercises | 105 (25.3) | 7.0 (6.0–9.0) |
| Relaxing activities | 101 (24.4) | 8.0 (6.0–9.0) |
| Turn SP off | 36 (8.7) | 7.0 (5.0–10.0) |
| Turn SP on | 133 (32.1) | 9.0 (8.0–10.0) |
| Change SP setting | 35 (8.5) | 7.0 (6.0–9.5) |
| *Change volume* * | 21 (5.1) | |
| *Change program* * | 16 (3.9) | |
| *Change microphone sensitivity* * | 4 (1.0) | |
| *Activate Forward focus* * | 1 (0.2) | |
| *Use Bluetooth devices* * | 1 (0.2) | |
| Other strategy/technique | 96 (16.7) | 7.0 (5.0–9.0) |
| *Ignore tinnitus* * | 35 (8.5) | |
| *Distractions* * | 16 (3.9) | |
| *Listen or create sounds/music* * | 13 (3.1) | |
| *Avoid and manage stress* * | 4 (1.0) | |
| *Rest* * | 4 (1.0) | |
| *Physical movement/position* * | 4 (1.0) | |
| *Relaxation activities* * | 3 (0.7) | |
| *Always wear SP* * | 2 (0.5) | |
| *Wait for it to go away* | 2 (0.5) | |
| *Turn tinnitus into music* * | 1 (0.2) | |
| *Avoid noisy situation* * | 1 (0.2) | |
| *Turn SP and hearing aid on* * | 1 (0.2) | |
| *Turn SP off* * | 1 (0.2) | |
| *Change SP settings* * | 1 (0.2) | |
| *Drug-based treatment* * | 1 (0.2) | |
| *"Strategies do not work"* * | 3 (0.7) | |
| No strategy/technique | 78 (18.9) | |
| **Night management strategies/techniques** | | |
| Listen to sound (without SP) | 28 (6.8) | 6.0 (5.0–7.0) |
| Listen to sound (with SP) | 32 (7.7) | 7.5 (5.8–9.3) |
| Relaxing activities | 81 (19.6) | 7.0 (6.0–9.0) |
| Wear the SP while sleeping | 12 (2.9) | 8.0 (6.0–10.0) |
| Other | 103 (24.9) | 6.0 (5.0–8.0) |
| *Ignore it* * | 19 (4.6) | |
| *Read* * | 16 (3.9) | |
| *Mental distraction* * | 14 (3.4) | |
| *Breathing exercises* * | 8 (1.9) | |
| *Listen to music* * | 6 (1.5) | |
| *Take medicines* * | 6 (1.5) | |
| *Turn SP on* * | 4 (1.0) | |
| *Physical exercises* * | 2 (0.5) | |
| *Watch TV* * | 2 (0.5) | |
| *Wait to be very tired to sleep* * | 3 (0.7) | |
| *No technique/strategy* * | 18 (4.3) | |
| No strategy/technique | 193 (46.6) | |
| **Tinnitus treatment (previous and current)** | | |
| Psychological treatment | 32 (7.7) | 6.0 (5.0–8.0) |
| Sound-based treatment | 13 (3.1) | 6.0 (5.0–7.0) |

*(Continued)*

**Table 6.** (Continued)

| Tinnitus management | N (%) | Median (IQR) |
|---|---|---|
| Drug-based treatment | 41 (9.9) | 6.0 (5.0–7.0) |
| Alternative therapies | 27 (6.5) | 5.0 (5.0–7.5) |
| Cochlear implant fitting session with an audiologist | 58 (14.0) | 7.0 (5.0–9.0) |
| Other | 32 (7.7) | 5.5 (5.0–8.0) |
| _Drug-based treatment_ * | 4 (1.0) | |
| _Relaxation therapies (mindfulness, sophrology)_ * | 3 (0.7) | |
| _Infusion_ * | 2 (0.5) | |
| _Sound-based therapy_ * | 1 (0.2) | |
| _Osteopathy_ * | 1 (0.2) | |
| _Hypnosis_ * | 1 (0.2) | |
| _Hyperbaric oxygen therapy_ * | 1 (0.2) | |
| _Self-performed strategies_ * | 4 (1.0) | |
| _"Tried everything"_ * | 1 (0.2) | |
| _No treatment_ * | 14 (3.4) | |
| No treatment | 285 (68.8) | |

IQR: Interquartile; N: Number of CI recipients; SP: Sound processor.

Tinnitus management level was rated according to the scale: 0 Very easy– 1–2–3–4–5–6–7–8–9–10 Impossible.

Each strategy/technique/treatment was rated according to the scale: 0 Worsens– 1–2–3–4–5 No change– 6–7–8–9–10 Improves.

Options marked with an asterisk (*) are extracted from an open field question and grouped into similar themes.

Many recipients did not use tinnitus management strategies at night (46.6% ($n = 193$); described in open field text: 4.3% ($n = 18$)). Among those who had used management strategies at night, 19.6% ($n = 81$) did relaxing activities and 16% ($n = 66$) listened to sound (with SP: 6.8% ($n = 28$); without SP: 7.7% ($n = 32$); described in open field text: 1.5% ($n = 6$)). Despite the manufacturer contra-indications, a few participants reported wearing their sound processor while sleeping and that this led to tinnitus improvement (2.9%, $n = 12$).

Less than a third of participants had tried a treatment provided by healthcare professionals (31.2%, $n = 129$). CI fitting session with an audiologist was the most frequently reported treatment (14.0% ($n = 58$)) and was also rated the most effective (7.0 (IQR: 5.0–9.0)). Additionally, 10.9% ($n = 45$) had drug-based treatment (9.9% ($n = 41$); described in open field text: 1.0% ($n = 4$)), 7.7% ($n = 32$) had psychological treatment and 3.3% ($n = 14$) tried sound-based treatment (3.1% ($n = 13$); described in open field text: 0.2% ($n = 1$)). Alternative therapies such as homeopathy, supplements, acupuncture, osteopathy were on average reported as not effective, with a median of 5.

## Discussion

In this mixed-method study, we explored the impact of tinnitus on adult CI recipients, situations impacting tinnitus, difficulties associated with tinnitus and their management strategies. The data collected from 136 CI users using a web-based forum discussion showed that tinnitus can affect everyday life of CI users in various ways and highlighted the heterogeneity in their tinnitus experiences. Four themes emerged from the thematic analysis: _tinnitus experience_, _situations impacting tinnitus_, _difficulties associated with tinnitus_ and _tinnitus management strategies_. We then developed and conducted a survey in 414 adult CI recipients experiencing tinnitus to assess the themes and sub-themes found in the forum study. While most participants experienced tinnitus independently of the sound processor use, tinnitus was on average perceived as not a problem when wearing the sound processor and a moderate problem when

not wearing it. The data collected in the survey highlighted specific situations, difficulties, and management tinnitus strategies, which were often dependent on the sound processor use. The study results highlight the need for further work to explore how to address tinnitus-related difficulties identified and how future studies should adapt how they assess tinnitus in CI recipients.

The discussion forum and survey outcomes highlighted the heterogeneity in tinnitus impact on everyday life of CI recipients. Different degrees of tinnitus awareness and annoyance were reported by participants, ranging from *aware but not bothered* to *always bothered* in the discussion forum. This heterogeneity in tinnitus annoyance has also been raised by studies analyzing large databases of CI users, where tinnitus was not a relevant problem in more than 70% of CI users with tinnitus, a moderate problem in around 20% of CI users with tinnitus and a severe or worse problem is less than 10% of CI users [10,15]. Participants did not all report the same difficulties as some had no difficulties associated with tinnitus where others reported several. These differences show that the distress associated with tinnitus is complex and patient specific. Therefore, tinnitus impact cannot be summarized as one common experience in the adult CI population. When exploring tinnitus heterogeneity, it has been suggested by Beukes et al. that subgroups based on tinnitus severity levels should be considered and managed differently [46]. A similar approach could be adopted by clinicians to identify CI users suffering from tinnitus and address their specific needs and associated difficulties.

Most participants experienced tinnitus independently of the sound processor use, but still a third reported having tinnitus only when not wearing the sound processor. This observation outlines the suppressive effect on tinnitus of electrical stimulation provided when the sound processor is worn in a third of the participants. However, this effect seems to be patient specific as the other 70% of participants did not report total suppression of tinnitus when wearing their sound processor. Previous studies have shown that electrical stimulation can still reduce tinnitus impact in CI recipients even if it does not completely suppress their tinnitus [10,18,19]. In the current survey, this was assessed using two questions distinguishing tinnitus impact when wearing the sound processor and while not wearing it. Most participants perceived their tinnitus as not a problem or a small problem when wearing their sound processor and as a moderate or big problem without their sound processor. When asked about tinnitus management strategies, one third of recipients reported turning on their sound processor during the day and even a few wore it while sleeping to better manage their tinnitus. Furthermore, difficulties related to tinnitus seem to intensify when not wearing the sound processor. Although not often present according to the low ratings given by the participants, all the difficulties assessed in the survey were significantly more present when not wearing the sound processor. This highlights the limitation of current tinnitus questionnaires, which do not distinguish the two conditions, with or without sound processor. Further work is required to reflect on how future studies including CI recipients should adapt how tinnitus impact is assessed in relation to the status of the sound processor.

The negative impact of hearing tests and CI programming sessions on tinnitus has been clearly reported by many CI recipients. From another perspective, Pierzycki et al. found that 80% of audiologists and 45% of CI recipients reported a negative effect of tinnitus on CI programming, making the programming sessions more difficult and tiresome [47]. They suggested that tinnitus may interfere with the process of refining the CI fitting, mainly because patients may confuse the presented stimuli with their tinnitus, which may challenge the accuracy of the threshold levels set during fitting. By the result of our study, one could reason that the sounds or stimuli presented during hearing tests and CI programming sessions increase tinnitus which could limit the process of CI fitting. This finding is of clinical importance because hearing tests and CI programming sessions remain two essential steps of the CI

rehabilitation in order to improve speech perception outcome, the primary intended aim of cochlear implantation. Further research is required to understand to what extent this impacts hearing outcomes of CI recipients with tinnitus. Nonetheless, when asked about treatment options, recipients rated CI fitting sessions with an audiologist as the most effective option to manage their tinnitus. Further work is needed to better understand how fitting can be optimized both for speech perception and tinnitus reduction and create guidelines for clinical application.

Interestingly, some sub-themes emerging from the forum discussion about tinnitus-related difficulties in CI users coincides with the items from current validated tinnitus impact questionnaires, but not all. Indeed, responses to the open field question in the survey revealed the presence of comorbidities being caused or aggravated by tinnitus in CI recipients (S3 Table), such as depressive feelings. In line with our findings, Basso et al. suggested that depression can have a negative effect to hearing-related difficulties and tinnitus impact in the general population experiencing tinnitus [48]. Further work is needed to understand the association and causal relationships between the comorbidities mentioned and tinnitus-related distress in CI recipients. The presence or impact of comorbidities is not assessed by the validated tinnitus questionnaires, whereas they seem to be associated with higher distress in tinnitus patients [46,48,49]. In current clinical practice, asking about comorbidities is not part of diagnostic standards, such as stated by the NICE guidelines on tinnitus developed in 2020 [50]. This emphasizes the need for further exploration of essential measurements and diagnostic tools to capture tinnitus-related comorbidities.

It is important to note that 19% of recipients in our studies reported not using any self-performed tinnitus management strategy during the day and 51% not using management strategies at night. Similarly, most participants did not try treatment provided by professionals. Based on the survey question assessing the impact of tinnitus, tinnitus was perceived as a big to very big problem by 28% of recipients in general, by 10.2% of recipients when wearing the sound processor and by 35.2% when not wearing it. These results emphasize that only a small proportion of recipients are seeking help for their tinnitus [10]. Targeting the population still suffering from tinnitus should be a priority to understand and address their needs.

A limitation of the study is that CI recipients suffering from tinnitus might be more inclined to participate in the survey than recipients having tinnitus with minor distress, involving a selection bias. Another limitation in interpreting the study results is a possible recall bias when asking about perceived changes in tinnitus since implantation. The study was restricted to cochlear implants recipients with a Cochlear Nucleus implant (Cochlear Ltd., Macquarie University, NSW, Australia) and therefore the study population represents only a selection of cochlear implant recipients. However, we do not expect the results with other implant types to be significantly different. Participants presented with different hearing profiles and device configurations, having one or two implants and for some wearing a hearing aid in the other ear. Based on our study, we do not know whether this would have affected the results; future studies may aim to find out. Although participants were instructed to focus only on the difficulties caused by tinnitus, independently of difficulties caused by hearing loss, hearing-related difficulties, and psychological problems reported may be due to their associated hearing loss. It can be acknowledged that it is hard to distinguish the difficulties related to the combination of tinnitus and hearing loss. This is also a limitation for our study and for all studies investigating tinnitus impact in CI patients where tinnitus and hearing loss constantly interact with similar factors. The survey was not developed following the COSMIN guidelines [51], mainly because the reliability of the survey was not evaluated. This was beyond the scope of the survey designed in the current study.

To our knowledge, no CI-specific survey has yet been developed to assess the impact of tinnitus on CI recipients in the literature. The mixed-method approach used in our study addressed the gap and identifies the difficulties of CI recipients experiencing tinnitus. The large sample size from six countries around the world depicts a diverse population representative of a typical population of CI recipients with tinnitus. The study provides evidence on the complexity of tinnitus associated with sound processor use and uncovers difficulties and situations associated with tinnitus that are exclusive to CI recipients. The complexity of tinnitus in CI recipients is often not fully addressed by clinicians by fear of unmanageable expectations. Given the findings of our mixed-method study, difficulties and complaints associated with tinnitus should be better identified and understood by clinicians in order to be addressed efficiently.

## Conclusion

This study explored the impact of tinnitus on CI recipients. Based on a qualitative analysis, a survey was developed to quantify the items identified (tinnitus presence, tinnitus impact, situations impacting tinnitus, difficulties associated with tinnitus, relationship with CI use, and management strategies). These findings provide insight in the potential benefits of the sound processor use and therefore intracochlear electrical stimulation on tinnitus impact. Clinicians and industry should focus on the identified difficulties to improve the condition of current and new CI recipients experiencing tinnitus post-implantation.

## Supporting information

**S1 Table. Structure of the developed survey.**
(PDF)

**S2 Table. Themes, sub-themes, and codes from the thematic analysis.**
(PDF)

**S3 Table. Comorbidities caused or aggravated by tinnitus.** All comorbidities are extracted from an open field question and grouped into similar themes.
(PDF)

**S1 Dataset.**
(ZIP)

## Acknowledgments

We would like to thank to Jochen Nicolai, Li Ma and Payal Sadrarangani with their help in managing the forum and the survey on the Cochlear Conversation platform.

## Author Contributions

**Conceptualization:** Kelly K. S. Assouly, Maryam Shabbir, Bas van Dijk, Derek J. Hoare, Inge Stegeman, Adriana L. Smit.

**Data curation:** Kelly K. S. Assouly, Maryam Shabbir.

**Formal analysis:** Kelly K. S. Assouly, Maryam Shabbir.

**Investigation:** Kelly K. S. Assouly, Maryam Shabbir.

**Methodology:** Kelly K. S. Assouly, Maryam Shabbir, Bas van Dijk, Derek J. Hoare, Inge Stegeman, Adriana L. Smit.

**Project administration:** Kelly K. S. Assouly.

**Resources:** Bas van Dijk, Michael A. Akeroyd, Robert J. Stokroos.

**Supervision:** Bas van Dijk, Derek J. Hoare, Michael A. Akeroyd, Robert J. Stokroos, Inge Stegeman, Adriana L. Smit.

**Validation:** Kelly K. S. Assouly.

**Visualization:** Kelly K. S. Assouly, Maryam Shabbir.

**Writing – original draft:** Kelly K. S. Assouly.

**Writing – review & editing:** Kelly K. S. Assouly, Maryam Shabbir, Bas van Dijk, Derek J. Hoare, Michael A. Akeroyd, Robert J. Stokroos, Inge Stegeman, Adriana L. Smit.

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
