## [Decision Letter · Decision Letter 0]

20 Jan 2023

PONE-D-22-35092The impact of tinnitus on adult cochlear implant recipients: a mixed-method approachPLOS ONE

Dear Dr. Assouly,

Thank you for submitting your manuscript to PLOS ONE. After careful consideration, we feel that it has merit but does not fully meet PLOS ONE’s publication criteria as it currently stands. Therefore, we invite you to submit a revised version of the manuscript that addresses the points raised during the review process.

We look forward to receiving your revised manuscript.

Kind regards,

Prashanth Prabhu

Academic Editor

PLOS ONE

2. In the competing interests statement within the manuscript and in the online submission form, please declare your affiliation with Cochlear Technology Centre and thoroughly report any potential competing interests related to this affiliation.

“I have read the journal's policy and the authors of this manuscript have the following competing interests: KKSA and MS received funding from the European Union’s Horizon 2020 research and innovation program under the Marie Sklodowska-Curie grant (agreement number 764604). KSSA and BvD are employed at Cochlear Technology Centre, Mechelen, Belgium. The content of the study belongs to the authors alone and do not reflect Cochlear Technology Centre policy. No further conflict of interest is reported by the authors.”

Reviewers' comments:

Reviewer's Responses to Questions

**Comments to the Author**

1. Is the manuscript technically sound, and do the data support the conclusions?

Reviewer #1: Partly

Reviewer #2: Yes

2. Has the statistical analysis been performed appropriately and rigorously? 

Reviewer #1: No

Reviewer #2: Yes

3. Have the authors made all data underlying the findings in their manuscript fully available?

Reviewer #1: Yes

Reviewer #2: Yes

4. Is the manuscript presented in an intelligible fashion and written in standard English?

Reviewer #1: Yes

Reviewer #2: Yes

5. Review Comments to the Author

Reviewer #1: Thank you for the opportunity to review this paper. The subject is of interest, where CI users experience with tinnitus is being explored.

The following aspects require clarification/ modifications;

General

1. The conclusion in the abstract should talk about key findings and not generic aspects

2. The secondary aim of looking at prevalence is not matching with the study design and is more of a profiling . This must be clarified.

3. Since the study is aligned to the use of a platform related to a particular company, how was bias avoided? what measures were taken?

Method

1. The design is mixed methods, with a qualitative and quantitative component. But the format of a forum discussion is not clear. Is there any reference for this method? how is it conducted when used as a research tool?

2. The qualitative study needs a theoretical framework, what guidelines were used to conduct and report? e.g. COREQ?

3. What were the interview / discussion guides? probes?

4. What was the level of training received by the interviewers/ discussion moderators

5. How was the data coded? manually? or software? need detailed description for these

6. Was deductive or inductive approach used to analyse the themes?

7. ICF is a framework that could have been considered since it is aligned to the aspects being explored in the study. Why were so many other questionnaires used instead? rationale is required

8. Were the subjects who participated in survey and forum mutually exclusive? or overlapping?

to what extent?

9. Data analysis is written in future tense

10. Statistical analysis: Validation of new tool (survey) should be using factor analysis, PCA or discriminant values. Cognitive interviews are not suited as primary validation methods.

Results

1. The results of the qualitative aspect is unusual with only quotes. Normally, the results are summarised and only few quotes are mentioned as examples. This is not in a readable format.

2. The results are extensive as it covers two almost independent aspects of the study. This does not do justice to either parts of the study in the current format.

This paper can be split into two independent papers to do justice to each part with adequate details on method and results.

Reviewer #2: Review for:

The impact of tinnitus on adult cochlear implant recipients: a mixed-method approach

Submitted to PLOS ONE

Manuscript Number: PONE-D-22-35092

Comments to the associate editor

Thank you for the opportunity to review this article. The motivation for the study is convincing and has good clinical implication. The paper has good writing. I do believe that if revised with minor changes such a paper will add value to the existing literature. With this intent, my review here attempts to highlight some of the key issues and indicate the errors that are reflected in the writing. I sincerely hope that the authors find this helpful in refining their manuscript.

Summary

The objective of this study was to evaluate the impact of tinnitus in CI recipients. The authors have derived four themes based on discussion in a web-based online forum. A survey was developed under tinnitus experience, situations impacting tinnitus, difficulties associated with tinnitus and tinnitus management. The results showed that tinnitus was of moderate degree with no sound processor and tinnitus was not a problem with sound processor.

Overall Impression (General Comments)

The strength of the manuscript is the motivation of the study and the flow in writing. The introduction section is adequate. The participants, materials and procedures are explained well. However, the main drawback of the study is the small number of participants used for validation of the survey questionnaire.

Detailed review

• Title: Adequate

Abstract:

• Well written

Main script:

Introduction:

o Adequate

Method:

• The sentence in page no. 7 ‘The number of questions on the sub-themes to be included in the survey was determined based on the number of sub-themes distracted from the quantitative analysis’ is not clear.

• The sample size, other participant details such as age range, the device details etc may be given in the method section.

Results and discussion:

• In page no. 23, Line nos 440 and 441: It is mentioned that ‘Without sound processor, group conversation and hearing difficulties were the most frequently reported, with a median score of 4, followed by difficulties in listening to radio or TV, concentration difficulties and stress, with a median score of 3’. The hearing and listening difficulties are expected to be poorer without the sound processor even in the absence of tinnitus. Authors have given this as a limitation of the study. However, even other aspects (given in Table 5) like stress, anger, anxiety etc. are also expected to be more without the sound processor in the absence of tinnitus.

• In my opinion, the participant details should be given in the method section

• Page no. 32, Line no 573: ‘All these results does emphasize’ may be changed to ‘All these results do emphasize’

• Did all the participants own a CI from the same manufacturers?

• Was there an association between the type of speech processor and tinnitus impact?.

• Was the tinnitus always present in the implanted ear?

• What about the duration usage of sound processor?

Figures:

• Figure 1 is not clear and the font in Figure 2 is not visible.

6. PLOS authors have the option to publish the peer review history of their article (what does this mean?). If published, this will include your full peer review and any attached files.

Reviewer #1: No

Reviewer #2: No

---

## [Author Response · Author response to Decision Letter 0]

24 Feb 2023

Dear reviewers, 

First of all, we want to thank you for your feedback on our manuscript. We addressed all comments and responded to each point you raised below.

We look forward to your response. 

Yours sincerely,

On behalf of all authors,

Kelly Assouly

---- 

Reviewer 1

General

1. The conclusion in the abstract should talk about key findings and not generic aspects.

  We agree with your remark and changed the text accordingly: «The qualitative analysis showed that tinnitus can affect everyday life of CI recipients in various ways and highlighted the heterogeneity in their tinnitus experiences. The survey findings extended this to show that tinnitus impact, related difficulties, and management strategies often depend on sound processor use. This exploratory sequential mixed-method study provided a better understanding of the potential benefits of sound processor use, and thus of intracochlear electrical stimulation, on the impact of tinnitus.».

2. The secondary aim of looking at prevalence is not matching with the study design and is more of a profiling . This must be clarified.

  The secondary aim was to assess the presence of tinnitus with regards to the sound processor status in a cohort of cochlear implant users. This secondary aim is an exploratory objective that was defined based on the hypothesis that the status of the sound processor has an influence on the presence of tinnitus. This secondary objective relies on the answers to questions Q1 (Table A1). We clarified the secondary aim in the introduction section by changing one sentence: «To investigate the influence of the sound processor on tinnitus, a secondary aim was to assess the presence of tinnitus with regards to the sound processor status in this cohort of cochlear implant users.».

3. Since the study is aligned to the use of a platform related to a particular company, how was bias avoided? what measures were taken?

  We agree with your remark and added two sentences to highlight this limitation in the discussion paragraph: “The study was restricted to cochlear implants recipients with a Cochlear Nucleus implant (Cochlear Ltd., Macquarie University, NSW, Australia) and therefore the study population represents only a selection of cochlear implant recipients. However, we do not expect the results with other implant types to be significantly different.” 

Method

4. The design is mixed methods, with a qualitative and quantitative component. But the format of a forum discussion is not clear. Is there any reference for this method? how is it conducted when used as a research tool?

  We added one sentence to clarify the format of the forum discussion: “The forum discussion was a moderator-led online forum discussion where a moderator encouraged the discussion and participants discussed specific topics through posting a series of messages and commented on each other’s post.”. We added a reference to practical guidelines for qualitative research using online forums from Im et al. (Im & Chee, 2012). 

5. The qualitative study needs a theoretical framework, what guidelines were used to conduct and report? e.g. COREQ?

  We did not specify a theoretical framework a priori but the work was certainly influenced by the ESIT Framework (Genitsaridi, Hoare, Kypraios, & Hall, 2020). We added a sentence on the theoretical framework in the methods section: “Although a theoretical framework was not explicitly defined a priori, the authors considered the ESIT Framework of variables defining and characterizing tinnitus sub-phenotypes particularly relevant to the current work as it describes the high dimensionality of tinnitus heterogeneity”.

As this was a mixed-method approach study we have selected to report it according to the Mixed Methods Article Reporting Standards (MMARS) (Levitt et al., 2018). For the qualitative component we followed the qualitative methods described by Braun and Clarke (Braun & Clarke, 2006). 

6. What were the interview / discussion guides? probes?

  The interview probe has been added to the publicly available dataset. 

7. What was the level of training received by the interviewers/ discussion moderators

  We added the level of training of the discussion moderators for the forum discussion and interviewers for the cognitive interviews: “All moderators were trained by the researcher KKSA about the phenomenological approach and research objectives.” and “The interviewer, KKSA, had extensive experience of conducting interviews as part of her doctoral studies”.

8. How was the data coded? manually? or software? need detailed description for these

  The data were coded manually. We added this information in the methods section. 

9. Was deductive or inductive approach used to analyse the themes?

  We used an inductive approach to analyse the themes as described line 138. 

10. ICF is a framework that could have been considered since it is aligned to the aspects being explored in the study. Why were so many other questionnaires used instead? rationale is required

  We agree with your remark and recognize that we could have considered the ICF to extract questions for the development of the survey. The ICF is a useful framework but has only one code for tinnitus (b2400). We think tinnitus questionnaires were more aligned with the aspects explored in the study such as tinnitus impact on daily life, situations impacting tinnitus, tinnitus-related difficulties, and management strategies. 

11. Were the subjects who participated in survey and forum mutually exclusive? or overlapping?

to what extent?

  We invited all Cochlear Conversation members who received a Cochlear Nucleus implant to voluntarily participate in the discussion forum and the survey by sending two independent emails separated by a 4-month period. This allowed subjects to participate in both the discussion forum and the survey. The subjects who participated in survey and forum might be overlapping, but as their participation was anonymous, we cannot estimate the overlap.

We added two sentences in the methods section to clarify this information: “Participants who had already taken part in the discussion forum could also take part in the survey. Therefore, the survey population might, to some extent, overlap with the one of the discussion forum.”

12. Data analysis is written in future tense

  We agree with your remark and changed the text accordingly.

13. Statistical analysis: Validation of new tool (survey) should be using factor analysis, PCA or discriminant values. Cognitive interviews are not suited as primary validation methods.

  We agree with your remark, changed the title of the paragraph “Survey validation” to “Survey validity” and adapted the text. Indeed, cognitive interviews were used to assess the face validity of survey question interpretations and response processes (Ryan, Gannon-Slater, & Culbertson, 2012).

Results

14. The results of the qualitative aspect is unusual with only quotes. Normally, the results are summarised and only few quotes are mentioned as examples. This is not in a readable format.

  We understand either reporting approach is valid if quotes are used to extend the text rather than provide example per se. Given the format of our data, the option to integrate quotes into the text to illustrate each sub-theme in the results section was the better option for readability. The results of the qualitative analysis are summarised in Figure 2 and in S3 Table. We updated Figure 2 for more clarity and added S3 Table as a Supplementary material. 

15. The results are extensive as it covers two almost independent aspects of the study. This does not do justice to either parts of the study in the current format.

This paper can be split into two independent papers to do justice to each part with adequate details on method and results.

  We considered two independent manuscripts, but this risked detracting from it being an integration of data in a mixed-method study, which should be reported for completeness in the first instance as a single article. This will also ensure it reaches the same audience as suggested by Stange et al. 2006 (Stange, Crabtree, & Miller, 2006).

Reviewer 2

Overall Impression (General Comments)

The strength of the manuscript is the motivation of the study and the flow in writing. The introduction section is adequate. The participants, materials and procedures are explained well. However, the main drawback of the study is the small number of participants used for validation of the survey questionnaire.

Detailed review

Method:

1. The sentence in page no. 7 ‘The number of questions on the sub-themes to be included in the survey was determined based on the number of sub-themes distracted from the quantitative analysis’ is not clear.

  Apologies, this was a typographical error, corrected to read: “The number of questions in the survey was determined based on the number of sub-themes derived from the quantitative analysis.”.

2. The sample size, other participant details such as age range, the device details etc may be given in the method section.

  We agree with your remark and, as suggested by the MMARS, we placed the participant sample size and characteristics in the methods section.

Results and discussion:

3. In page no. 23, Line nos 440 and 441: It is mentioned that ‘Without sound processor, group conversation and hearing difficulties were the most frequently reported, with a median score of 4, followed by difficulties in listening to radio or TV, concentration difficulties and stress, with a median score of 3’. The hearing and listening difficulties are expected to be poorer without the sound processor even in the absence of tinnitus. Authors have given this as a limitation of the study. However, even other aspects (given in Table 5) like stress, anger, anxiety etc. are also expected to be more without the sound processor in the absence of tinnitus.

  We agree with your remark and changed the text accordingly in the discussion section: “Although participants were instructed to focus only on the difficulties caused by tinnitus, independently of difficulties caused by hearing loss, hearing-related difficulties and psychological problems reported may be due to their associated hearing loss.”

4. In my opinion, the participant details should be given in the method section

  We agree with your remark and, as suggested by the MMARS, we placed the participant sample size and characteristics in the methods section.

5. Page no. 32, Line no 573: ‘All these results does emphasize’ may be changed to ‘All these results do emphasize’

  We agree and have simplified to “These results emphasize”.

6. Did all the participants own a CI from the same manufacturers?

  All participants have a Cochlear Nucleus implant from Cochlear Limited as described in line 112, 129 and 205. 

7. Was there an association between the type of speech processor and tinnitus impact?

  We did not collect data on the type of sound processor of participants. Moreover, we did not assess the association between the type of sound processor and tinnitus impact as it was not within the scope of our study. Therefore, we cannot answer this question. 

8. Was the tinnitus always present in the implanted ear?

  We did not compare the side of implantation to the tinnitus side as it was not within the scope of our study. Therefore, we cannot answer this question.

9. What about the duration usage of sound processor?

  Participants reported using their sound processor on average 15 hours per day (IQR: 13 – 16 hours per day). This result indicates that most participants use their sound processor the whole day and are regular users. However, this information is difficult to interpret without more details about their hearing profile or without comparison with another group, patients without tinnitus for example.

Figures:

10. Figure 1 is not clear and the font in Figure 2 is not visible.

  We agree with your remark and modified the two figures. 

References:

Braun, V., & Clarke, V. (2006). Using thematic analysis in psychology. Qualitative Research in Psychology, 3(2), 77–101. Retrieved from https://doi.org/10.1191/1478088706qp063oa

Genitsaridi, E., Hoare, D. J., Kypraios, T., & Hall, D. A. (2020). A review and a framework of variables for defining and characterizing tinnitus subphenotypes. Brain Sciences, 10(12), 1–21. Retrieved from https://doi.org/10.3390/brainsci10120938

Im, E. O., & Chee, W. (2012). Practical guidelines for qualitative research using online forums. Computers, Informatics, Nursing : CIN, 30(11), 604–611. Retrieved from https://doi.org/10.1097/nxn.0b013e318266cade

Levitt, H. M., Bamberg, M., Creswell, J. W., Frost, D. M., Josselson, R., & Suárez-Orozco, C. (2018). Journal article reporting standards for qualitative primary, qualitative meta-analytic, and mixed methods research in psychology: The APA Publications and Communications Board task force report. The American Psychologist, 73(1), 26–46. Retrieved from https://doi.org/10.1037/amp0000151

Ryan, K., Gannon-Slater, N., & Culbertson, M. J. (2012). Improving Survey Methods With Cognitive Interviews in Small- and Medium-Scale Evaluations. American Journal of Evaluation, 33(3), 414–430. Retrieved from https://doi.org/10.1177/1098214012441499

Stange, K. C., Crabtree, B. F., & Miller, W. L. (2006). Publishing multimethod research. Annals of Family Medicine. United States. Retrieved from https://doi.org/10.1370/afm.615

---

## [Decision Letter · Decision Letter 1]

6 Apr 2023

The impact of tinnitus on adult cochlear implant recipients: a mixed-method approach

PONE-D-22-35092R1

Dear Dr. Assouly,

We’re pleased to inform you that your manuscript has been judged scientifically suitable for publication and will be formally accepted for publication once it meets all outstanding technical requirements.

Kind regards,

Prashanth Prabhu

Academic Editor

PLOS ONE

Additional Editor Comments (optional):

Reviewers' comments:

Reviewer's Responses to Questions

**Comments to the Author**

1. If the authors have adequately addressed your comments raised in a previous round of review and you feel that this manuscript is now acceptable for publication, you may indicate that here to bypass the “Comments to the Author” section, enter your conflict of interest statement in the “Confidential to Editor” section, and submit your "Accept" recommendation.

Reviewer #3: All comments have been addressed

2. Is the manuscript technically sound, and do the data support the conclusions?

Reviewer #3: Yes

3. Has the statistical analysis been performed appropriately and rigorously? 

Reviewer #3: Yes

4. Have the authors made all data underlying the findings in their manuscript fully available?

Reviewer #3: Yes

5. Is the manuscript presented in an intelligible fashion and written in standard English?

Reviewer #3: Yes

6. Review Comments to the Author

Reviewer #3: 1. There is a knowledge gap regarding the loudness and pitch matching of tinnitus pre and post tinnitus management.

2. Duration of tinnitus experienced by the patient isn't clearly mentioned.

3. Gender specific management outcome isn't explained clearly.

4. Point no. 71 , sentence formation can be improvised for more meaningful outcome.

5. Table 1, Demographic data , Hearing loss (H/L) type can also be written in this manner - (a) H/L in one ear, normal hearing sensitivity in other, (b) Symmetrical hearing loss, (c) Bilateral H/L (one ear affected more than other)

7. PLOS authors have the option to publish the peer review history of their article (what does this mean?). If published, this will include your full peer review and any attached files.

Reviewer #3: **Yes: **Archana Gupta

---

## [Editor Report · Acceptance letter]

12 Apr 2023

PONE-D-22-35092R1 

The impact of tinnitus on adult cochlear implant recipients: a mixed-method approach 

Dear Dr. Assouly:

I'm pleased to inform you that your manuscript has been deemed suitable for publication in PLOS ONE. Congratulations! Your manuscript is now with our production department. 

Kind regards, 

on behalf of

Dr. Prashanth Prabhu 

Academic Editor

PLOS ONE